# A Blue Light-Responsive Strong Synthetic Promoter Based on Rational Design in *Chlamydomonas reinhardtii*

**DOI:** 10.3390/ijms241914596

**Published:** 2023-09-27

**Authors:** Chen Chen, Jun Chen, Guangxi Wu, Liling Li, Zhangli Hu, Xiaozheng Li

**Affiliations:** Guangdong Technology Research Center for Marine Algal Bioengineering, Guangdong Provincial Key Laboratory for Plant Epigenetics, Shenzhen Engineering Laboratory for Marine Algal Biotechnology, Longhua Innovation Institute for Biotechnology, College of Life Sciences and Oceanography, Shenzhen University, Shenzhen 518060, China

**Keywords:** *Chlamydomonas reinhardtii*, synthetic promoter, blue light, cadaverine, rational design, synthetic biology

## Abstract

*Chlamydomonas reinhardtii* (*C. reinhardtii*) is a single-cell green alga that can be easily genetically manipulated. With its favorable characteristics of rapid growth, low cost, non-toxicity, and the ability for post-translational protein modification, *C. reinhardtii* has emerged as an attractive option for the biosynthesis of various valuable products. To enhance the expression level of exogenous genes and overcome the silencing of foreign genes by *C. reinhardtii*, synthetic promoters such as the chimeric promoter AR have been constructed and evaluated. In this study, a synthetic promoter GA was constructed by hybridizing core fragments from the natural promoters of the acyl carrier protein gene (*ACP2*) and the glutamate dehydrogenase gene (*GDH2*). The GA promoter exhibited a significant increase (7 times) in expressing GUS, over the AR promoter as positive control. The GA promoter also displayed a strong responsiveness to blue light (BL), where the GUS expression was doubled compared to the white light (WL) condition. The ability of the GA promoter was further tested in the expression of another exogenous cadA gene, responsible for catalyzing the decarboxylation of lysine to produce cadaverine. The cadaverine yield driven by the GA promoter was increased by 1–2 times under WL and 2–3 times under BL as compared to the AR promoter. This study obtained, for the first time, a blue light-responsive GDH2 minimal fragment in *C. reinhardtii*, which delivered a doubling effect under BL when used alone or in hybrid. Together with the strong GA synthetic promoter, this study offered useful tools of synthetic biology to the algal biotechnology field.

## 1. Introduction

*Chlamydomonas reinhardtii* (*C. reinhardtii*), a unicellular eukaryotic green alga, has long served as a preeminent model organism for exploring cilia structure and function, photosynthesis, and cell cycle regulation. Its distinctive structure, unique growth mode, and rapid growth rate make it an ideal candidate for such investigations [1,2]. In recent years, the utilization of microalgae as a “green cell factory” for the production of biofuel, food, and pharmaceuticals has emerged as a rapidly growing industry [3]. *C. reinhardtii*, with its eukaryotic characteristics of post-translational modification, has the capability to express and accumulate complex proteins with accurate folding and assembly [4,5], ensuring that recombinant proteins closely resemble their natural counterparts in structure and function [6,7]. Despite these advantages, *C. reinhardtii* encounters challenges when expressing exogenous genes. Integration of exogenous genes primarily occurs via non-homologous recombination, with a relatively low frequency of homologous recombination [8]. Such integration events are often accompanied by point deletions, rearrangements, and translocations at the recombination site [9]. Additionally, *C. reinhardtii* possesses a gene-silencing mechanism that can transcriptionally or post-transcriptionally inhibit the expression of exogenous genes [10,11]. Gene silencing serves as a protective mechanism against invading viruses, transposons, and exogenous genes, and while it may be inevitable [12,13], it poses obstacles to efficient gene expression in *C. reinhardtii*.

The precise regulation of gene expression primarily relies on promoters, which initiate and control gene transcription by binding transcription factors via cis-regulatory elements (CREs) within the promoter region [14,15]. Early studies focused on identifying natural strong promoters in *C. reinhardtii* or utilizing promoters from higher plants such as *Arabidopsis* and rice, but significant breakthroughs were not achieved. However, the field of synthetic biology has emerged as a new approach, with synthetic promoters becoming a prominent trend [16,17]. Synthetic promoters can overcome limitations associated with natural promoters, including restricted induction conditions and low transcription frequency and strength, thereby significantly enhancing the expression of exogenous genes [18,19,20,21]. Several endogenous promoters have been identified to promote exogenous gene expression, such as the promoters of the ribosomal small subunit RBCS2, heat shock protein HSP70A, and photosystem I protein PSAD [22,23]. The chimeric AR promoter, composed of promoters HSP70A and RBCS2, has been widely used but has demonstrated suboptimal performance in protein expression despite contributing to elevated transcriptional levels. Consequently, recent research has focused on identifying or synthesizing novel regulatory elements and assembling them into potent synthetic promoters capable of driving high transcription efficiency [24,25]. Synthetic promoters can be either created by top-down or bottom-up approaches [26]. The top-down approach is essentially the modification of the native and well-characterized promoters, by point mutations, insertion of CREs, truncation, and hybridization. On the contrary, the bottom-up approach is to generate completely new constructs from scratch, via the assembly of standardized blocks [27] or computational design followed by gene synthesis [28]. In one study, publicly available mRNA expression data were utilized to identify CREs from promoters of most highly expressed genes. The CREs were randomly inserted into nucleotide backbones to generate synthetic algal promoters (saps), among which seven exhibited superior performance in driving fluorescence protein expression compared to AR [29]. Another study uncovered a light-responsive CRE called sequences over-represented in light-repressed promoters (SORLIP) from the promoter sequence of the light-inducible protein gene (*LIP*) of *Dunaliella*. When two copies of SORLIPs were connected to the minimal *LIP* promoter, the resulting synthetic promoter exhibited a much stronger light inducibility than the original *LIP* promoter [30].

In this study, a top-down approach with stepwise truncation of native promoters and hybridization of minimal promoter fragments was used to construct a novel synthetic promoter, GA, in *C. reinhardtii* using rational design, enabling strong and robust expression of exogenous genes. The GA promoter exhibits strong inducibility in response to blue light (BL), further doubling the expression of exogenous genes. Compared to the commonly used AR promoters, the AG promoter demonstrated a remarkable 7-fold increase in GUS expression and a 1- to 2-fold increase in the production of the heterogeneous product, cadaverine under WL. Under BL illumination, the expression levels of both proteins further doubled. The development of this promoter offers a valuable tool for facilitating the expression of recombinant proteins in *C. reinhardtii*, thereby providing a powerful platform for genetic engineering applications in both academic research and industrial settings.

## 2. Results

### 2.1. Native Promoter Identification and Construction of Transgenic Lines

To generate synthetic promoters for *C. reinhardtii* with high strength and BL inducibility, RNA-seq transcriptomics was performed to examine gene expressions at the transcriptional level under both WL and BL conditions. Six genes were selected with fpkm > 500 and log2(Fold change) > 1 from pathways related to crucial cellular functions (Figure 1A). *PSAD* with Log2(Fold Change) of approximately −1 was included as a reference. *LHCBM3*, *FAP12*, *PETM,* and *ACP2* were identified as genes transcriptionally upregulated by BL, while *GDH2*, *PSAD,* and *FAP310* showed transcriptional downregulation specifically under BL (Figure 1B). These findings suggest the presence of blue light-responsive CREs within the promoter regions of these genes. The promoter sequence of the selected gene was determined as the region from the last nucleotide of the 3’ UTR of the upstream gene to the start codon of the selected gene. The length of all the promoters ranged from approximately 1000 to 2000 base pairs (Table 1).

To evaluate the strength of the above endogenous promoters in expressing exogenous genes, each promoter sequence was synthesized in connection with codon-optimized *GUS* gene sequence and cloned into the *pJ1DCF* vector, followed by the RBCS2 3′UTR to yield the *pJ1DCF-CreGUS* plasmids (Figure 2A). These plasmids were then transformed into the wild-type (WT) strain CC-849 to generate the *CreGUS*-expressing lines. The coding sequence of *CreGUS* is effectively regulated by the different promoters, while the RBCS2 3’-UTR element functions as the termination signal for the cessation of *CreGUS* transcription. An RBCS2 promoter drove the expression of the bleomycin resistance gene (Shble) as a selection marker. After transformation, cells were screened on agar plates supplemented with antibiotics for the isolation of positive colonies. The integration of the expression cassette into the nuclear genome of these colonies was verified by genomic PCR analysis using primers specific to the target genes (Figure 2B). For each transformation, different numbers of positive clones were obtained by the presence of bands of the correct sizes on the agarose gel.

### 2.2. Characterization of Native Promoters in CreGUS Expression

The strength of the six endogenous promoters was evaluated for their expression of *CreGUS* at both the transcriptional (Figure 3A) and translational levels (Figure 3B) together with AR and PSAD promoters used as positive controls. The results clearly demonstrate that the transcriptional activity of the GUS gene driven by the GDH2 promoter was significantly higher compared to all other promoters, exhibiting approximately a 10-fold increase over the AR promoter. Despite its high transcriptional activity, the protein expression level, as indicated by GUS activity, was among the lowest observed. The ACP2 promoter exhibited the opposite pattern. While its transcriptional activity showed no difference from the AR promoter, the protein expression level was more than doubled compared to the AR promoter. These findings sparked the straightforward idea to leverage the robust transcriptional strength of GDH2 and combine it with the strong translational capacity of the ACP2 promoter via linear hybridization.

### 2.3. Promoter Truncation and Characterization of Promoter Fragments

To locate functional regions within the lengthy native promoters, a stepwise truncation strategy was employed to generate shorter fragments harboring fewer CREs. Specifically, ACP2 and GDH2 promoters were truncated to generate ACP2-D1, ACP2-D2, ACP2-D3 and GDH2-D1 and GDH2-D2 (Figure 4A and Table 2). The length of each fragment was determined based on the distribution of CREs within the original promoters. Subsequently, these truncated fragments were ligated with the GUS gene, cloned into the *pJ1DCF* vector, and introduced into *C. reinhardtii*. The relative mRNA expression level of the GUS gene (Figure 4B) and GUS activity (Figure 4C) driven by each promoter fragment were measured, providing valuable insights into their regulatory capacities.

The impact of truncating the ACP2 promoter on gene expression was investigated, revealing substantial improvements at both the transcriptional and translational levels. ACP-D1, the shortest fragment, exhibited a 4-fold increase in GUS mRNA expression compared to the original promoter, accompanied by enhanced GUS activity. ACP-D2 displayed a similar trend, further confirming the positive effect of promoter truncation. However, ACP-D3 exhibited worse performance compared to the original promoter. In the case of GDH2, the truncation of the original promoter did not yield any improvement in gene expression. In fact, the relative mRNA expression levels of *GUS* were even lower for GDH2-D1 and GDH2-D2 compared to GDH2. Mixed effects were observed in GUS activity levels for the two GDH2 fragments.

To achieve the objective of designing synthetic strong promoters with BL inducibility, the strengths of the promoter fragments were assessed in BL or WL conditions, and the results were compared (Figure 5). For ACP2-D1 and ACP2-D2, the relative mRNA expression levels showed no significant difference between WL and BL. A significant disparity was observed for ACP2-D3 where the relative *GUS* expression level under BL significantly increased compared to the WL condition. For the GDH2 fragments, a clear response to BL was evident, as relative *GUS* expression levels approximately doubled under BL compared to WL conditions for both GDH2-D1 and GDH2-D2. In terms of GUS activity levels from transgenic lines, ACP2-D1 and ACP2-D2 showed the highest GUS activity with no significant difference between WL and BL conditions. While ACP2-D3 exhibited a dramatic increase in relative *GUS* expression under BL over WL conditions, the GUS activity of ACP2-D3 under BL was much lower than in WL conditions. The GUS activity of both GDH2 fragments showed a 1-time increase under BL over WL conditions. These observations strongly suggest GDH2-D1 and GDH2-D2 as potential regulatory elements for blue light-mediated gene expression.

### 2.4. Construction and Characterization of Chimeric Promoters

Taking into consideration the fragment length and gene expression at the transcription and translation levels, ACP2-D1, ACP-D2, GDH2-D1, and GDH2-D2 were selected as the building blocks of the synthetic promoter. The synthetic promoter was synthesized by connecting GDH2-D1, GDH2-D2, HSP70A, and PSAD with ACP-D1 in sequence, respectively (Table 2). The GUS activity of transgenic lines expressing the *GUS* gene under the control of these synthetic promoters was measured (Figure 6A). The GDH2-D1/ACP2-D1 construct showed the highest GUS activity, followed by GDH1-D2/ACP2-D1, HSP70A/ACP2-D1, and PSAD/ACP2-D1. The GUS activity of the GDH2-D1/ACP2-D1 construct averaged at 840 pmol 4-MU/min/μg protein was better than both of its constituents, GDH2-D1 and ACP2-D1 constructs with average GUS activity of 90 and 500 pmol 4-MU/min/μg protein. To examine the blue light-inducing effect of GDH2-D1 in the chimeric promoters, it was connected to PSAD, ACP2-D1, and ACP-D2, respectively (Table 2), and used to drive GUS expression under WL and BL conditions (Figure 6B). All three chimeric promoters demonstrated a doubling of GUS activity under BL compared to WL conditions, with GDH2-D1/ACP2-D1 still exhibiting the highest performance.

### 2.5. Production of Cadaverine Using GDH2-D1/ACP2-D1 Promoter

The chimeric promoter GDH2-D1/ACP2-D1 (GA) showed the greatest strength with strong BL inducibility in expressing GUS. It was then anticipated that this chimeric promoter would have the same effect on other heterogeneous genes. Cadaverine, a derivative of L-lysine, is commonly used as a building block for biopolyamide. It is synthesized in many eukaryotes via decarboxylation of L-lysine catalyzed by lysine decarboxylases. The lysine decarboxylase gene cadA from *E. coli* was transformed into CC-849 together with either AR or GA promoter. The cadaverine production was measured on HPLC using cell lysate of transgenic *C. reinhardtii* (Figure 7). The standard curve and spectra of each measurement were illustrated in Appendix A. Under WL, the AR construct yielded 0.22 ± 0.05 mg Cadaverine/g dcw (dry cell weight) while the AG construct yielded a more than double level of 0.51 ± 0.10 mg Cadaverine/g dcw. Under the BL condition, the cellular productivity of the AR construct was 0.33 ± 0.07 mg Cadaverine/g dcw, showing no significant difference from the WL condition, while the AG construct produced 0.79 ± 0.03 mg Cadaverine/g dcw, representing an approximately 60% increase over WL condition.

## 3. Discussion

The objective of this study was to develop a synthetic promoter in *C. reinhardtii* responsive to BL, addressing limitations associated with natural promoters in terms of transcription efficiency, intensity, and inducible conditions. Based on RNA transcriptomic information, six new native promoters were obtained from genes with differential expressions under WL and BL conditions (Figure 1B). GDH2 and ACP2 promoters were found to have the greatest transcription and translation activities among all promoters tested, including the most commonly used AR and PSAD promoters (Figure 3). A relatively large deviation was observed within the ACP2 group (*n* = 6) for GUS expression (Figure 3B), most likely due to positional effects from random insertions of exogenous genes. Pooling many transformants and measuring the total population is an option to reduce noise from positional effects but this does not give information on expression range. To reduce the influence of positional effects, constructs with large deviations in GUS expression were generated and measured twice in separate experiments. For example, the ACP2 construct was generated and measured twice as illustrated in Figure 3B and Figure 4C, and comparable results were obtained.

A stepwise truncation strategy was then utilized to shorten the lengthy regions upstream of the transcription start site (TSS) of the native promoters based on predictions of potential functional CREs (Figure 4A). Subsequently, the strength of each promoter fragment was examined by measuring the expression of the *GUS* gene or GUS protein in transgenic cells (Figure 4B,C). Enhanced transcription and translation activities were observed in ACP2 and its fragments, with ACP2-D1 having the highest activities. GDH2 and its fragments experienced mixed results, and while GDH2-D2 showed the lowest gene expression level, its protein expression was the highest compared to GDH2 and GDH2-D1. This could be attributed to the mixed effects of CREs of different functions that made the behaviors of promoters somehow unpredictable. Furthermore, GDH2-D1 showed strong responsiveness to BL (Figure 5), as both relative gene expression and GUS activity levels increased by 1-fold. By connecting GDH2-D1 with ACP2-D1, a chimeric promoter GA was constructed that exhibited an even stronger expression of GUS protein in *C. reinhardtii*. The GUS activity from the GA construct was increased by more than 7-fold than the AR promoter (Figure 6A). Under BL, the GUS activity was further doubled (Figure 6B). The enhanced strength of the GA promoter was also tested to express another exogenous gene cadA for the production of cadaverine. Again, the GA promoter showed superior performance over the AR promoter with a doubling effect under BL compared to the WL condition (Figure 7). These findings highlight the broad applicability and effectiveness of this synthetic promoter, providing a valuable tool for future studies in *C. reinhardtii* by overcoming limitations related to insufficient expression.

Although the study focused on the design of a synthetic promoter to enhance gene expression at the translation levels in *C. reinhardtii*, our design strategy holds great potential to inspire future research. Traditional approaches to synthetic promoter design often relied on utilizing CREs from native promoters, involving their characterization, mutation, repetition, and rearrangement, followed by synthesis of the promoter for functional testing in host organisms such as plants or protoplasts [31]. In the present context, the design of synthetic promoters typically involves four key steps: characterization of CREs, promoter design, construction of reporter modules driven by synthetic promoters, and functional testing of the synthetic promoters. Identifying CREs and predicting their functionality are the initial and most challenging aspects of such research. Advances in high-throughput molecular biology techniques have facilitated genomics-based approaches to study gene expression [32,33,34]. Databases such as the Synthetic Promoter Libraries (SPLs) and large-scale analyses of promoter expression profiles have aided in the prediction and analysis of CREs in promoters [35,36].

The stepwise truncation strategy deployed in the study did not rely on precise identification and knowledge of CREs associated with the sequences of native promoters. The long sequences of original promoters were truncated stepwise by 200–400 bp to form D3, D2, and D1 fragments (Figure 4A and Table 2). The D1 fragments of around 100 bp primarily comprise TSS and minimum CREs. The D2 and D3 then comprise more CREs including MYB, CAAT, and G box, according to software predictions. Interestingly, simple truncation of the promoter led to significant improvements in transcription and protein expression levels (Figure 4). This may be attributed to the removal of non-essential elements or silencers [37] during truncation, enabling more efficient binding of trans-acting factors to the region and facilitating alterations in spatial conformation, thereby enhancing transcription and translation. By hybridizing the two D1 fragments of GDH2 (138 bp) and ACP2 (120 bp) promoter, a chimeric promoter GA (258 bp) was generated, exhibiting the strong and robust expression of two exogenous genes *GUS* and *cadA* multiple times higher than the AR counterpart. The relatively short length of the GA promoter ensures great feasibility and flexibility in metabolic engineering applications, as it occupies minimal space and exerts minimal impact on the original gene when employed [21,27,38,39].

In *C. reinhardtii*, an array of inducible promoters with different strengths and regulatory mechanisms were developed [10]. These promoters can be triggered by controlled environmental cues and deliver a rapid, dose-dependent response [11]. The addition of chemicals [40], depletion of nutrients [41], application of heat shock [13], and irradiation of high light [30] and green light [42] are some examples of induction systems. Among them, light is advantageous over others as it avoids chemical toxicity and nutrition scarcity, is instantaneously effective compared to heat shock, and its use of a specific light spectrum as an inducer is particularly attractive because it ensures a more specific control on the gene expression. To date, no blue light-responsive genetic part has been developed in *C. reinhardtii*. Although the exact sequence of the blue light-responsive motif associated with GDH2-D1 was not identified in the study, the 138-bp GDH2-D1 served as a good minimal fragment that can be hybridized to yield synthetic promoters with strong and consistent BL inducibility (Figure 6B).

In conclusion, the synthetic GA promoter with strong expression capacity, BL inducibility, and relatively short length offers a great tool for the research and development of *C. reinhardtii* in terms of genetic manipulation and metabolic engineering, boosting their potential for biotechnology and synthetic biology.

## 4. Materials and Methods

### 4.1. C. reinhardtii Culture

The cell wall-deficient wild-type *C. reinhardtii* strain CC-849 was obtained from the Chlamydomonas Resource Centre (https://www.chlamycollection.org/) and routinely cultured in Tris–acetate–phosphate (TAP) medium under continuous light (60 μmol/m^2^/s) at 25 °C. *E. coli* DH5α cells were used for normal DNA manipulation and cultured in LB medium at 37 °C.

### 4.2. Plasmid Construction

The nucleotide sequences of promoters were obtained from Phytozome13, available online at https://phytozome-next.jgi.doe.gov/info/Creinhardtii_v5_6 (accessed on 12 December 2020; Appendix A). The coding sequence of the β-glucuronidase (GUS) gene was obtained as previously reported with codon modifications [43]. The promoter-GUS sequences were synthesized by GENERAL BIO (Chuzhou, China). The coding sequence of the *E. coli*
*cadA* gene was obtained for nuclear expression as previously reported [44]. The promoter-cadA sequences were synthesized by GENERAL BIO. The expression vector used in the study was pJ1DCF containing a bleomycin resistance gene cassette driven by an RBCS2 promoter. Each promoter-GUS or promoter-cadA sequence was cloned into BglII and KpnI restriction sites of pJ1DCF to form the GUS reporter plasmids or cadaverine expression plasmids (Appendix A).

### 4.3. C. reinhardtii Transformation and Transgenic Strains

The vector was transformed into CC-849 by glass bead aggregation. Prior to transformation, vectors pJ1DCF-GUSs or pJ1DCF-cadAs were linearized by NotI digestion. Briefly, for each construct, CC-849 was grown to the mid-log phase (1–2 × 10^6^ cells/mL), and then cells were harvested and concentrated to 2 × 10^8^ cells/mL. Next, 300 μL of the cell suspension, 0.3 g of glass beads (0.4–0.6 mm), 30 μL of PEG 6000 (50% *w*/*v*), and 3 μg of linearized vector were mixed in a 1.5-mL centrifuge tube and vortexed at the maximum speed for 20 s. Cells recovered for 18–20 h in 10 mL of TAP (dark/weak light) were then plated onto two TAP plates supplemented with 8–10 μg/mL zeocin. A total of 40 randomly selected colonies grown up in 1–2 weeks were picked and transferred into a TAP medium to propagate for validations. The positive transformants were verified by amplifying either the *GUS* or *cadA* gene with approximately 2000 bp in length, using primers F-ATGCTGCGCCCCGTGGAGA and R-TTACTGCTTGCCGCCCTGCT for GUS, and F-ATGAACGTTATTGCAATATTGAATCACA and R-CTTTCAATACCTTAACGGTATAGCG for *cadA*. Ten randomly selected positive transformants for each construct was used for subsequent characterizations. If the number of positive clones was less than 10, then all positive clones were picked.

### 4.4. RNA Isolation and qRT-PCR

The total RNA was isolated during the phase of exponential growth using the SteadyPure Quick RNA Extraction Kit (Accurate Biology, AG21023, Changsha, China). Subsequently, an assessment of RNA integrity was conducted employing the NanoDrop2000 Ultra Microscope Photometer (Thermo, Waltham, MA, USA). The synthesis of first-strand cDNA was executed utilizing the HiScript II 1st Strand cDNA Synthesis Kit (Vazyme, R211, Nanjing, China), strictly adhering to the instructions by the manufacturer. To analyze patterns of gene expression, quantitative real-time polymerase chain reaction (qRT-PCR) was performed using the SYBR Green Master kit (Roche, 4913914001, Basel, Switzerland) in accordance with guidelines by the manual on an ABI QuantStudio 6 Flex Detection Device (Thermo, Waltham, MA, USA). Each experimental condition comprised three distinct biological repetitions and an additional three technical repetitions. The *α-tubulin* gene (*TUA1*, CHLRE_03g190950v5) was employed as an internal reference for normalization, while the relative expression levels of the target genes were deduced using the 2 −∆∆Ct method.

### 4.5. GUS Activity Quantitative Assay

The quantitative assay was performed as described by Jefferson et al. [45] with slight modifications. *C. reinhardtii* cells were cultivated for 7 days. Approximately 100 mg (fresh weight) was collected from each sample and frozen in liquid nitrogen. The frozen samples were homogenized with GUS extraction buffer for GUS fluorometric assays (50 mM phosphate buffer at pH 7.0, 10 mM, EDTA, 0.1% Triton X-100, 0.1% SDS, and 10 mM β-mercaptoethanol) and lysed for 5 min after adding Cell Culture Lysis Reagent (Promega). Subsequently, the samples were centrifuged at 12,000× *g* and 4 °C for 2 min. The supernatant was used for measurement of the protein concentration, which was determined using the PiercetTM BCA Protein Assay Kit (Thermo Scientific) with serum albumin as the standard. An amount of 15 μg of the protein extract was used, which was mixed with assay buffer containing 1 mM 4-methylumbelliferyl-b-glucuronide (4-MUG) (Sigma Aldrich, St. Louis, MO, USA) at 37 °C. The reaction was terminated by the addition of 200 mM Na_2_CO_3_ to a final concentration of 180 mM. GUS activity was measured by determining fluorescence with Varioskan Flash Multimode Microplate Reader (Thermo Scientific) at an excitation wavelength of 355 nm and an emission wavelength of 460 nm. The standard curve was created using 4-methylumbelliferone (4-MU) (Sigma Aldrich, St. Louis, MO, USA) as a substrate. The GUS activity was expressed as picomoles of 4-MU per microgram of protein per minute.

### 4.6. Promoter Regulatory Sequence Analysis and Promoter Fragments

The regulatory sequence of each promoter was analyzed using various programs such as PLACE, PlantCARE, and PlantPAN 3.0 databases for the identification of putative cis-acting regulatory elements and the Berkeley Drosophila Genome Project (BDGP) and YAPP eukaryotic core promoter predictor for the localization of core promoter region. The transcription initiation sites were determined via 5′RACE-PCR using SMARTer RACE 5′/3′ Kit (Takara Bio, San Jose, CA, USA) according to the manufacturer’s instructions. Promoter fragments were generated by serial deletions and were named ACP2-D1 (−65 to +55), ACP2-D2 (−418 to +55), ACP2-D3 (−701 to +55), GDH2-D1 (−14 to +124), and GDH2-D2 (−189 to +124). For further analysis, fragments were combined generating six chimeric promoters as follows: GDH2-D1/ACP-D1, GDH2-D2/ACP-D1, PSAD/ACP-D1, HSP70/ACP-D1, GDH2-D1/PSAD, GDH2-D1/ACP-D2 and GDH2-D2/ACP-D2. Each chimeric promoter was linked to the GUS gene and individually cloned into a pJ1DCF vector to generate each GUS reporting plasmid (Appendix A), which was later transformed into wild-type CC-849 cells.

### 4.7. Cadaverine Measurement

Transgenic *C. reinhardtii* cells harboring the *cadA* gene from three randomly selected positive clones were grown in a TAP medium and harvested at the log phase. Then, 40 mg cells were mixed with 0.2 mL of 5% trichloroacetic acid. and vortexed for 1 h. The mixture was centrifuged at 12,000× *g* rpm and 4 °C for 10 min. An amount of 100 µL supernatant was mixed with 30 µL of saturated sodium bicarbonate solution and 200 µL of 6 mg/mL dansyl chloride-acetone solution and vortexed thoroughly. The resulting mixture was put into a water bath at 40 °C for 45 min in the darkness. An amount of 16 µL of 25% ammonia solution was added to remove excess dansyl chloride. The solution was once again vortexed and centrifuged at 12,000× *g* rpm and 4 °C for 10 min. The resulting supernatant was transferred into a 250 µL vial for subsequent analysis.

Cadaverine measurement was carried out using a Waters 2695 HPLC with a Waters 2475 fluorescence detector and a Waters C18 column (Milford, MA, USA). The excitation and emission wavelengths were 340 nm and 515 nm, respectively. The column temperature was 40 °C. The mobile phase was methanol-water at 1 mL/min. The gradient elution operation was shown in Table 3.

### 4.8. RNA-Seq Transcriptomics

The total RNA was extracted from CC-849 cells grown at either WL or BL (60 μmol/m^2^/s) in TAP medium for 5 days using the RNAprep Pure Plant Plus Kit (TIANGEN, Beijing, China). Sequencing libraries were generated using the NEBNext Ultra RNA Library Prep Kit for Illumina (NEB, Ipswich, MA, USA) following the manufacturer’s recommendations, and sequenced on an Illumina NovaSeq 6000 platform with an S4 module kit (Illumina, San Diego, CA, USA). DESeq2 [46] was used to detect DEGs with the filters fpkm > 500, and log2(Fold change) > 1.

### 4.9. Statistical Analysis

To examine the statistical significance between experimental groups, a two-sided Student’s t-test assuming non-homologous variances was conducted to compare means from at least three independent experimental replicates, with significance determined at 0.05 (*) or 0.01 (**).

## Figures and Tables

**Figure 1 ijms-24-14596-f001:**
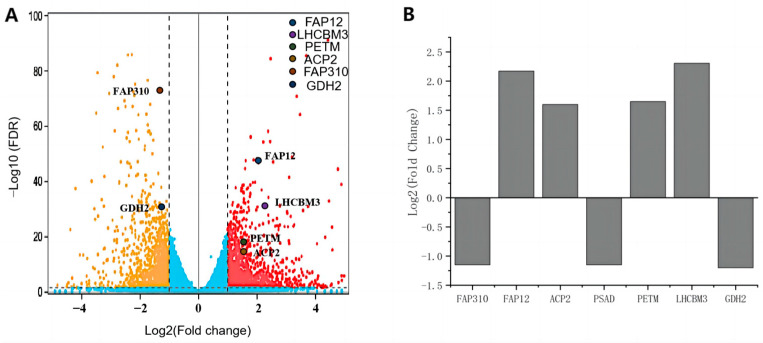
Selected endogenous genes showing differential expressions under white and blue light. (**A**) Volcanic plots of differentially expressed genes (DEGs) in *C. reinhardtii* cells cultivated under white light or blue light. FDR: adjusted *p* value. DEGs selection criteria: fpkm > 500 and log2(Fold change) > 1. (**B**) Differential expression levels of the six selected genes. PSAD was included as a reference. PETM: cytochrome b6f complex PetM subunit. FAP12: flagellar associated protein 12. LHCBM3: chlorophyll a-b binding protein. ACP2: acyl-carrier protein. GDH2: glutamate dehydrogenase. FAP310: flagellar associated protein 310.

**Figure 2 ijms-24-14596-f002:**
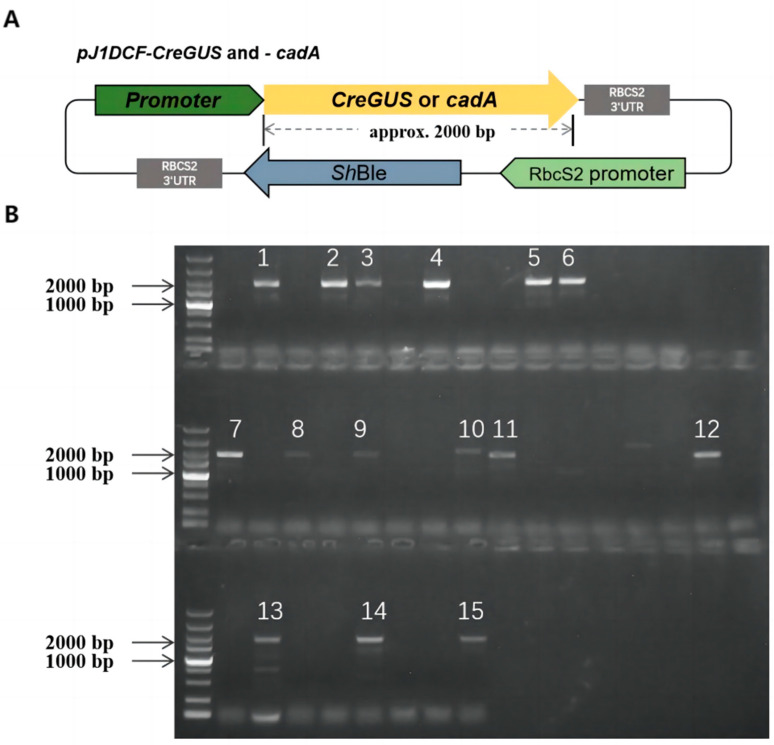
Construction and validation of *CreGUS*-expressing *C. reinhardtii* transgenic lines. (**A**) Schematic representation of the *pJ1DCF-CreGUS* or *-cadA* vector. Vector elements were labeled. *CrGUS*: β-glucuronidase gene with codon optimization. *cadA*: *E. coli* lysine decarboxylase gene. ShBle: bleomycin resistance gene from *Streptoalloteichus hindustanus*. Primer locations and the size of the corresponding amplicon used to verify positive transformants were indicated by the black and grey lines. (**B**) Representative agarose gel electrophoresis showing if *CreGUS* or *cadA* gene was integrated into *C. reinhardtii* genome. Bands were amplicon obtained by PCR amplification of *CreGUS* or *cadA* gene from clone genomic DNA. The gel showed 15 positive transformants from 40 randomly selected colonies of AR-GUS construct. For each transformation, 40 colonies were randomly selected for positive verification. Ten positive transformants were then randomly selected for subsequent characterizations as the number of positive clones was larger than 10.

**Figure 3 ijms-24-14596-f003:**
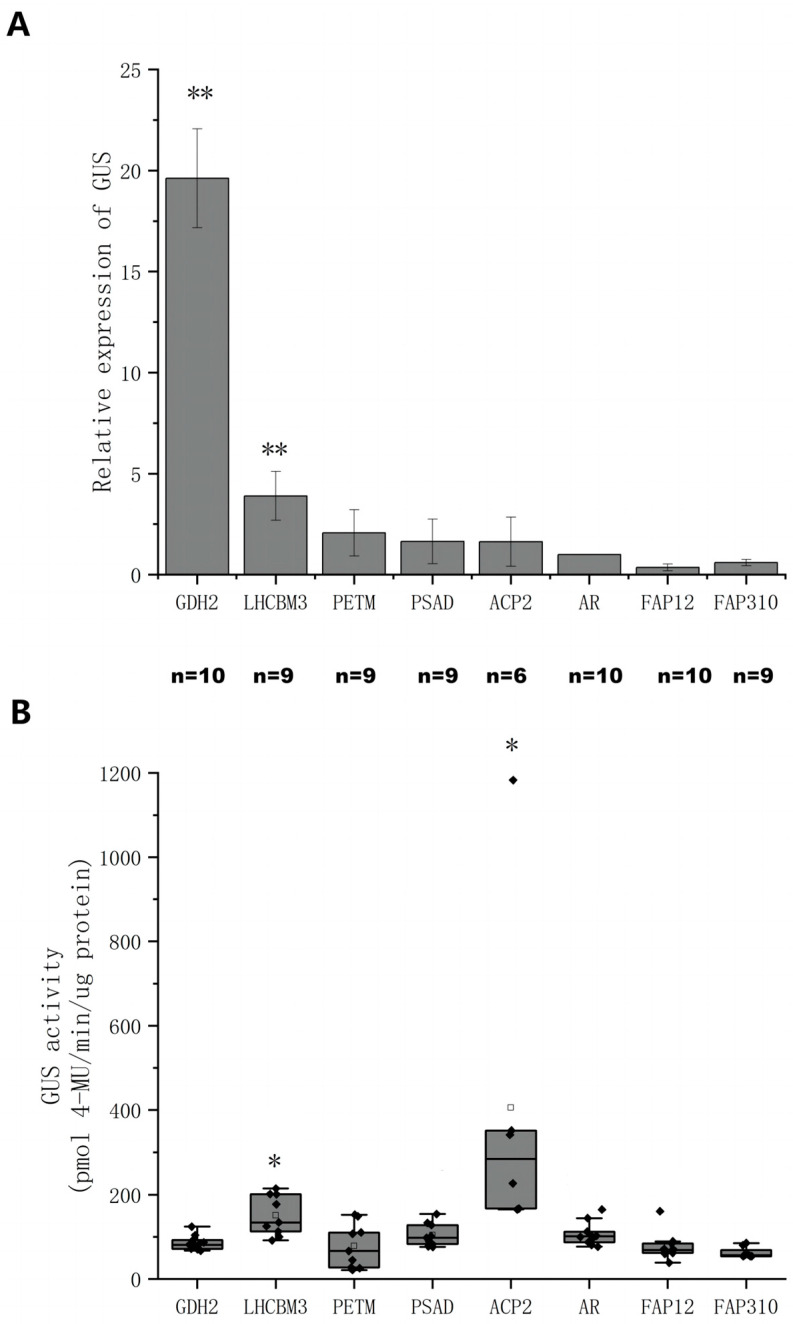
Quantitative analysis of GUS expressions under the control of different endogenous promoters. (**A**) The relative GUS gene expression levels driven by different promoters relative to AR. Results were presented as an average of three replicates with standard deviation. (**B**) Box plots of GUS activities from cell lysate of different *C. reinhardtii* transgenic lines. Black dots were individual data points. The square was the average of each experimental group with *n* specified, respectively. ** and * indicate statistical significance as compared to the AR construct.

**Figure 4 ijms-24-14596-f004:**
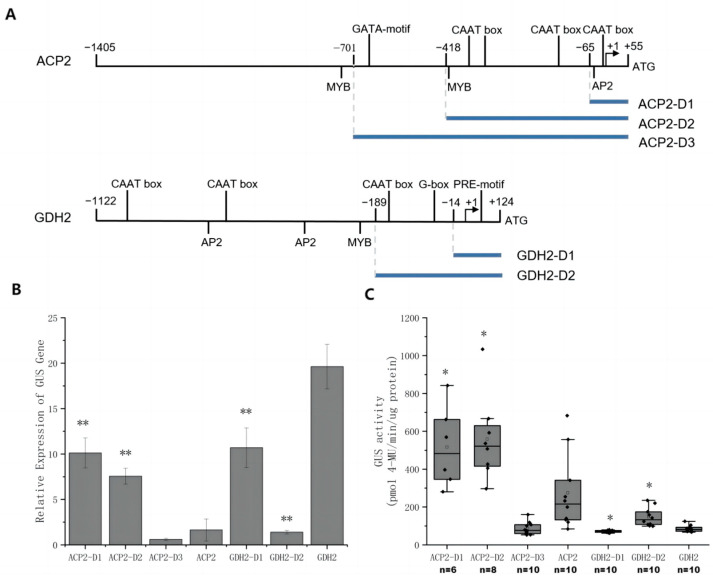
Native promoter truncation and characterization of promoter fragments. (**A**) Illustration of promoter truncation of ACP2 and GDH2 promoter. The original promoter sequence was labeled with predicted cis-regulatory elements (CREs) using various programs such as PLACE, PlantCARE, and PlantPAN 3.0 databases. The promoter fragments were designated as D1, D2, or D3. Arrows represent transcription start site predicted by 5′RACE. (**B**) The relative GUS gene expression levels driven by different promoter fragments relative to AR construct. Results were presented as an average of at least three replicates with standard deviation. (**C**) Box plots of GUS activities of cell lysate from different *C. reinhardtii* transgenic lines. Black dots were individual data points. The square was the average of each experimental group with *n* specified, respectively. ** and * indicate statistical significance as compared to the original promoter.

**Figure 5 ijms-24-14596-f005:**
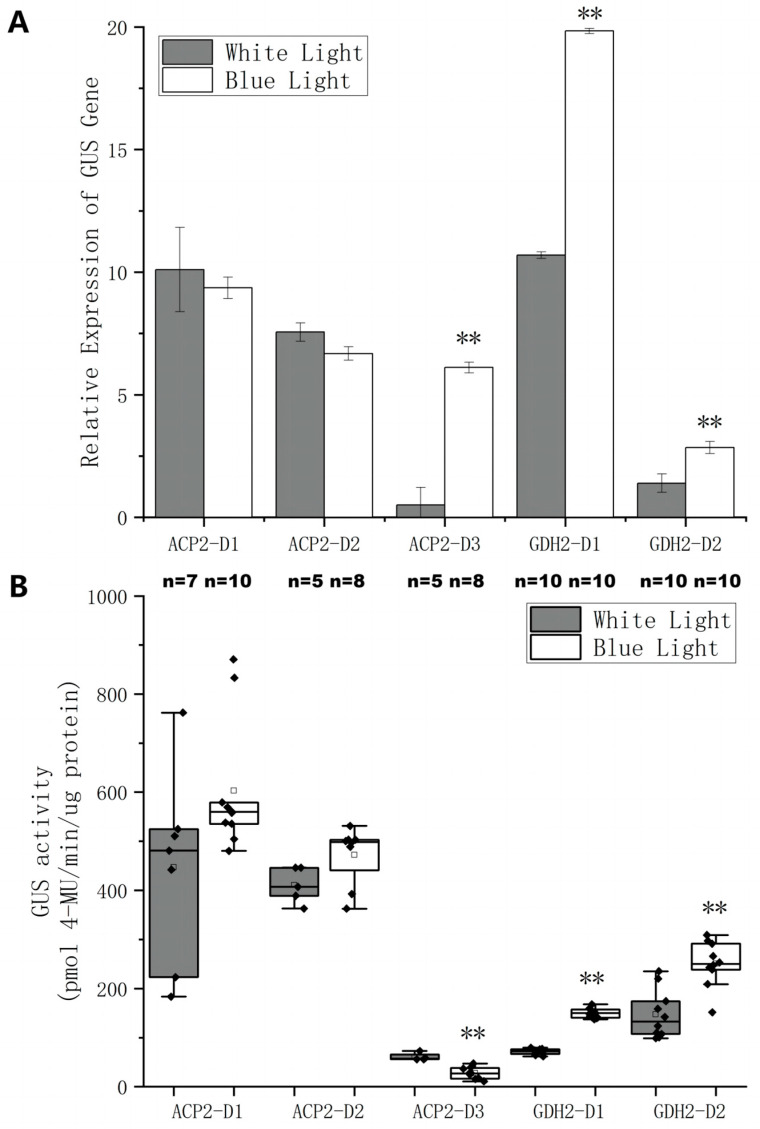
Quantitative analysis of GUS expressions under the control of different promoter fragments in *C. reinhardtii* cells grown at white light or blue light. (**A**) The relative GUS gene expression levels driven by different promoter fragments in cells grown at white light or blue light relative to AR construct. Results were presented as an average of at least three replicates with standard deviation. (**B**) Box plots of GUS activities from cell lysate of different *C. reinhardtii* transgenic lines grown at white light or blue light. Black dots were individual data points. The square was the average of each experimental group with *n* specified, respectively. ** indicates statistical significance as compared to white light condition.

**Figure 6 ijms-24-14596-f006:**
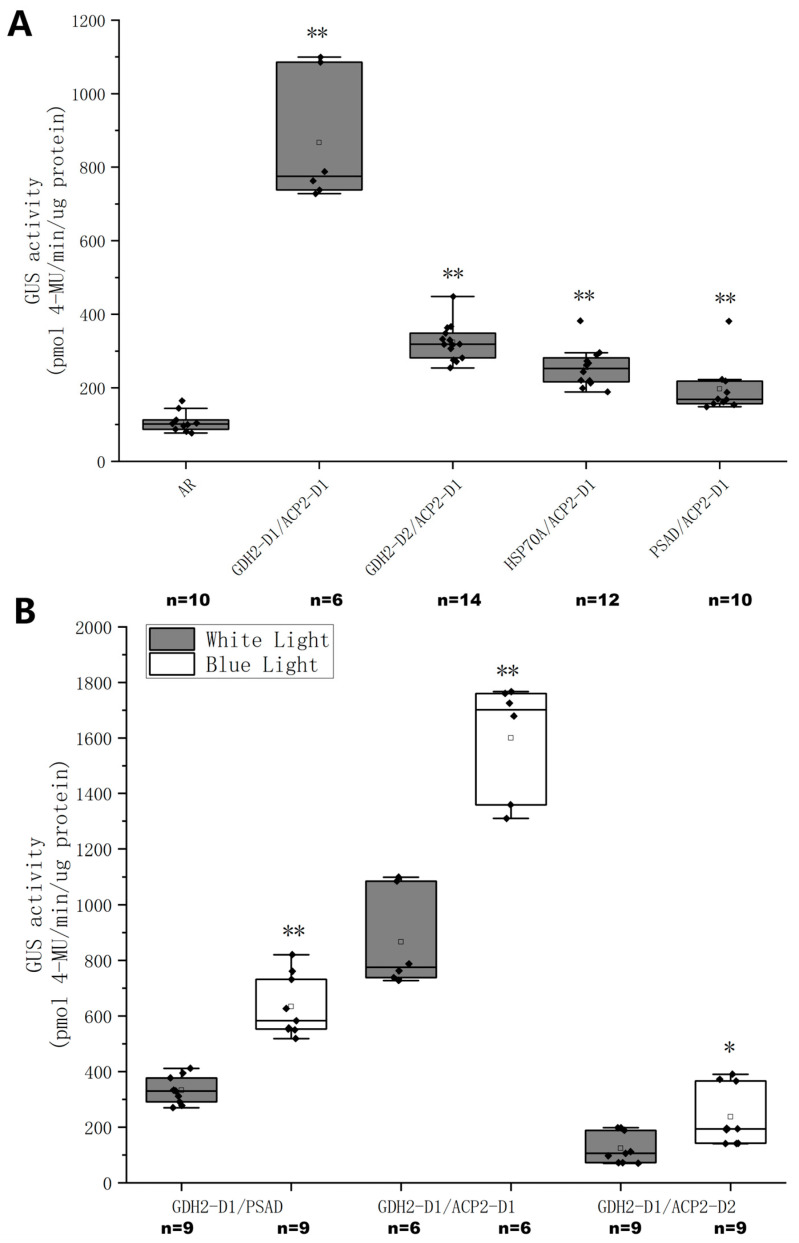
Quantitative analysis of GUS expressions under the control of different chimeric promoters in cells grown at white light or blue light. (**A**) Box plots of GUS activities from cell lysate of different *C. reinhardtii* transgenic lines grown at white light. ** indicates statistical significance as compared to AR. (**B**) Box plots of GUS activities from cells grown at white light and blue light. Black dots were individual data points. ** and * indicate statistical significance as compared to white light condition. The square was the average of each experimental group with *n* specified, respectively.

**Figure 7 ijms-24-14596-f007:**
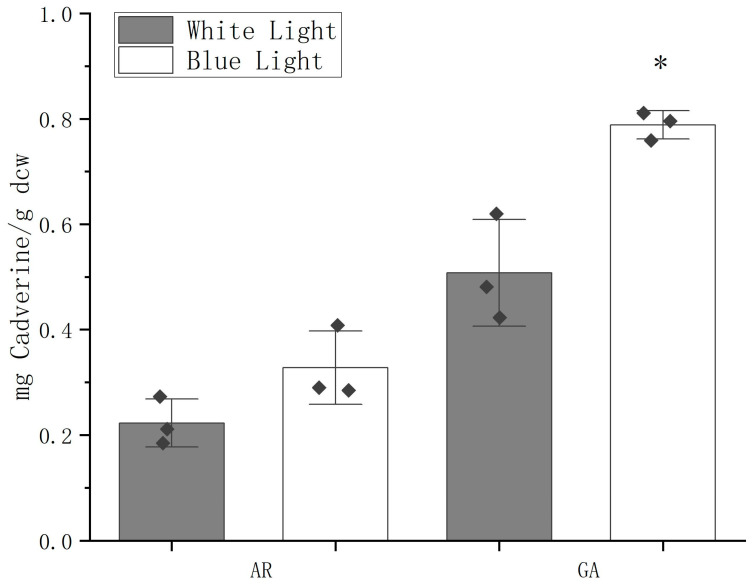
Cadaverine production from transgenic *C. reinhardtii* cells expressing cadA driven by either AR or GA promoters under white light and blue light conditions. Results are presented as average. Error bars represent the standard deviation of results from three randomly selected positive clones. * indicates statistical significance as compared to white light condition.

**Table 1 ijms-24-14596-t001:** Information on the native promoter used in the study.

Gene Code	Promoter Length (bp)	Name	Symbol
CHLRE_02g077800v5	1083	Flagellar Associated Protein 310	FAP310
CHLRE_09g390615v5	2143	Flagellar Associated Protein 12	FAP12
	461	Hybrid of HSP70A and RBCS2 promoter from *C. reinhardtii*	AR
CHLRE_13g577100v5	1460	Acyl-carrier protein 2	ACP2
CHLRE_05g238332v5	822	Photosystem I reaction center subunit II, 20 kDa	PSAD
CHLRE_12g546150v5	1088	Chloroplast cytochrome b6f PetM subunit	PETM
CHLRE_04g232104v5	1542	Light-harvesting complex II chlorophyll a/b binding protein M3	LHCBM3
CHLRE_05g232150v5	1246	Glutamate dehydrogenase	GDH2

**Table 2 ijms-24-14596-t002:** Information on the synthetic promoters used in the study.

Promoter Name	Length (bp)
ACP2-D1	120
ACP2-D2	473
ACP2-D3	756
GDH2-D1	138
GDH2-D2	213
GDH2-D1 + ACP2-D1 (GA)	258
GDH2-D2 + ACP2-D1	333
HSP70A + ACP2-D1	383
PSAD + ACP2-D1	942
GDH2-D1 + PSAD	960
GDH2-D1 + ACP2-D2	611

**Table 3 ijms-24-14596-t003:** The detail of gradient elution operation for measuring cadaverine concentration using HPLC. The mobile phase was made up of A (water) and B (methanol) at ratios as specified.

Time (min)	A% (Water)	B% (Methanol)
0	45	55
3	38	62
5	32	68
9	15	85
12	12	88
13	0	100
15	0	100
17	45	55
19	45	55

## Data Availability

The RNA transcriptomic data is available upon reasonable request to the corresponding author. All other data presented in this study are available within the article.

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
