# Peer review of "A Blue Light-Responsive Strong Synthetic Promoter Based on Rational Design in Chlamydomonas reinhardtii"

_ijms, 2023, doi:10.3390/ijms241914596_

Round 1
Reviewer 1 Report
Summary
The single-celled green alga Chlamydomonas reinhardtii is a popular choice for biosynthesis due to its rapid growth, cost-effectiveness, and ability for post-translational protein modification. The authors have developed synthetic promoters to boost exogenous gene expression and counteract gene silencing in this alga. The GA promoter, created by merging fragments from the acyl carrier protein gene (ACP2) and the glutamate dehydrogenase gene (GDH2), was found to be highly effective. It expressed the GUS gene seven times more effectively than the AR promoter and responded strongly to blue light, doubling the GUS expression compared to white light. Furthermore, the GA promoter enhanced the expression of the exogenous cadA gene, responsible for cadaverine production. Under blue light, the cadaverine yield driven by the GA promoter was 2-3 times higher than that driven by the AR promoter. This study has introduced a blue light-responsive GDH2 fragment in C. reinhardtii, which provides valuable synthetic biology tools for algal biotechnology.
Major points
1. I strongly advise improving the quality (resolution) of all figures.
2. What were the approximate numbers of transformant colonies for each construct in sections 2.1, 2.2, 2.3, and 2.4? Were they roughly similar across all constructs?
3. Position effects significantly influence transgene expression in Chlamy. How were position effects accounted for in this paper, especially in sections 2.2, 2.3, and 2.4? Were multiple transformants pooled for the measurements of promoter intensity/cadaverine production? If so, how many?
4. Another comment on position effects: Section 2.5 mentions results "from three randomly selected positive clones." This sample size is too small to compare promoters effectively. Other studies, such as those found at doi.org/10.1038/s41467-020-19983-4 and doi.org/10.1128/ec.00055-07, typically use several hundred transformants to mitigate this phenomenon.
5. Figure 2B is somewhat redundant as it does not contribute substantial value to the manuscript, serving mainly as an example of genotyping the transformants. Moreover, it lacks descriptions of the clones. The figure is only referenced in the main text if the citation in line 128 refers to Figure 2B. The panel would be more informative if it clarified what each lane represents and how the study further characterized these transformants.
6. The discussion is coherent, but an important point is absent. How do the cadA expression results compare to those from other researchers (e.g., reference 44 by Freudenberg et al.) when using the GA and AR promoters? Is there higher or lower cadaverine production?
7. In Figure 1B, the authors present differential expression levels of six genes, comparing white light to blue light regarding fold change. However, there are seven genes, including PSAD. The reason for its inclusion needs to be clarified. If PSAD is not a control, the authors should consider adding the expression change of at least one housekeeping gene for comparison.
8. They do not specify the approximate number of Chlamy transformants used to evaluate all promoter activities. While they mention picking colonies from the transformation plates, they should specify the count, especially since they do not discuss other methods to address positional effects despite acknowledging random integration.
9. Fig 4,5, Why APC2-derived mutants show a correlation between GUS expression and activity, but GDH2-derived mutant promoters show an opposite correlation?
10. They need to analyze or discuss why GDH2-D2 has lower expression than GDH2-D1 and GDH2.
Minor points:
1. There seems to be some confusion about the gene nomenclature of the GDH2 promoter and its fragments as in several occasions, such as in line 162, line 176, line 204, line 408 it is referred to as “GDHA” or “GDHA-D1”. This should be thoroughly revised.
2. In line 351, “(1–2×106 cells/mL)“, the number “6” and in line 352 “2×108 cells/mL”, the number “8” should be written in superscript.
3. In line 325/326, either “its potential” or “their potential” should be removed.
4. In line 326, there should be only one dot at the end of the sentence.
5. In figure 1, panel B, I suppose the authors mean “PSAD” instead of “PASD”.
6. If this will not be done by the editors later, I would advise to revise the materials and methods section for spaces between numerical values and unit symbols, such as in line 353, 385, 390, 416 etc.
Reviewer 2 Report
The article by Chen and colleagues describes a study aimed at constructing a strong promoter for the expression of exogenous genes in Chlamydomonas reinhardtii. The unicellular green alga C. reinhardtii is of growing industrial interest for the production of biomolecules. The discovery of stronger promoters than those already available is therefore of great interest. The use of light to induce expression is an added advantage in this organism, and the authors selected their constructs for induction by blue light (BL). They thus obtained the first BL-inducible promoters.
In a first step, the authors identified by comparative RNAseq 6 genes whose expression was stimulated after BL exposure. From the 5' intergenic regions of these genes, the authors isolated 6 promoters whose strength was assessed for RNA synthesis and translation of the GUS reporter gene. The results of these 2 tests showed improvements in efficiency, but also that transcription efficiency was not necessarily corroborated by translation efficiency.
Through a succession of deletions and ligations, the authors constructed several hybrid promoters which were tested and compared under the same conditions.
The result is a GA promoter that outperforms the reference AR promoter in transcription and translation of the GUS reporter gene under blue light induction conditions. However, in another in vivo cadaverine production assay, the increase in expression was only twofold compared with the standard AR promoter.
The experimental procedure is rational and adequately explained. The manuscript text is clearly written. A discussion of the site of integration by recombination could be added and would improve the article. In C. reinhardtii, exogenous genes are introduced mainly by non-homologous recombination, which raises questions about the effect of integration on the strength of the promoters constructed by the authors. Several clones of each construct were obtained and the clones of interest were identified by PCR. However, the integration context was not studied. Did the authors compare the efficacy of several clones of the same construct? Is it conceivable that different integration sites lead to varying efficiencies depending on genomic context? The authors should discuss this point in the light of existing literature in the field.
Minor points:
In the abstract, the authors define the GA promoter but mention the AR promoter without any explanation. It should be mentioned here that this is a chimeric reference promoter.
Legend Figure 2, part B: specify that these are fragments obtained by PCR amplification from clone genomic DNA.
Figure 4A: font size too small
Figure 6B: Is it possible to add the values of the control AR promoter?
In a few places in figure 7 and the corresponding text, the authors use cadverine instead of cadaverine.
Reviewer 3 Report
The manuscript entitled “A Blue Light-Responsive Strong Synthetic Promoter based on Rational Design in Chlamydomonas reinhardtii” by Chen and coworkers describes the design of an strong light inducible promoter, which moreover display a larger enhancement, upon exposure of cells to blue light. The obtained synthetic promoter, called GA, is shown to have higher expression level of a common reporter gene (GUS) and also result in a higher production yield a proof-of-concept compound (cadaverine) with respect to the reference synthetic promoter employed in Chlamydomonas, called AR.
The study is well organised and the main rational of the experimental approaches for the choice of candidate light-responsive promoter, generation of minimal responsive element and design of a synthetic hybrid are sufficiently well presented and the resulting observation are clearly discussed. Henceforth, the shown results appear to be reliable and the newly synthetised GA promoter appears to have effectively an application potential for obtaining light-inducible synthesis of compounds in Chlamydomonas. I could, then, not envisage any major issue in the study which I would therefore recommend for publication.
The only possible point of weakness relates to the absence of a clear identification of the elements responsible for dampening/silencing transcription in the originally identified on basis of nature response, only the broad regions are identified in search for “minimal constituents” to be then assembled in the hybrid promoter(s) that were obtained by progressive cropping the “natural” ones. This is briefly discussed by the authors, and it seems agreeable that is not the main task of the study although it represents a key issue underlying the functionality of the synthetic promoter obtained in the study.
However, the authors mention that the “cropping” was designed on the basis of previous pre-informatic screening of the natural promoters. I would then suggest that, at least, the putative silencing elements preset in cropped region shall be presented, eventually in the SI and briefly discussed.
Round 2
Reviewer 1 Report
The authors reply to all my comments.